# Effects of Digital Neurohabilitation on Attention and Memory in Patients with a Diagnosis of Pediatric Obesity: Case Series

**DOI:** 10.3390/brainsci15040353

**Published:** 2025-03-28

**Authors:** Noemí Cárdenas-Rodríguez, Claudia Andrea Chávez-Mejía, Vania Sofía Gardida-Álvarez, Norma Angélica Labra-Ruíz, Julieta Griselda Mendoza-Torreblanca, Eduardo Espinosa-Garamendi

**Affiliations:** 1Laboratorio de Neurociencias, Subdirección de Medicina Experimental, Instituto Nacional de Pediatría, Mexico City 04530, Mexico; noemicr2001@yahoo.com.mx (N.C.-R.); norma_labra@yahoo.com.mx (N.A.L.-R.); julietamt14@hotmail.com (J.G.M.-T.); 2Fundación Cognitive Habilitation, Mexico City 03100, Mexico; 3Departamento de Neuropsicología del Desarrollo y Neurohabilitación, Clínica Cognition, Mexico City 03100, Mexico; claudia@cognition.mx; 4Escuela de Psicología, Universidad Panamericana, Mexico City 03920, Mexico; vaniagardida0509@gmail.com; 5Unidad de Neurohabilitación y Conducta, Servicio de Neurología, Dirección Médica, Instituto Nacional de Pediatría, Mexico City 04530, Mexico

**Keywords:** digital neurohabilitation, attention, memory, pediatric obesity

## Abstract

**Objective:** Obesity represents a health risk and several studies have linked this clinical entity to cognitive deficits. Among the neuropsychological rehabilitation tools, Peak, a digital application, has shown positive results as a therapeutic method. The aim of this work was to measure, for the first time, cognitive deficits and the effects of Peak digital cognitive neurohabilitation therapy in patients diagnosed with obesity. **Methods:** Peak treatment was offered to the parents who agreed and lasted 6 months, including the neurocognitive evaluation. The patients used Peak five times a day for 20 min. The Neuropsychological Attention and Memory Battery (NEUROPSI) was applied before and after the intervention. **Results:** The results revealed posttest changes in attention and executive function, memory, and total attention and memory. Significant clinical changes were observed, and the diagnostic range increased from severe to moderate. **Conclusions:** We concluded that, through an intervention with the Peak app, it is possible to enable attention and memory, which represent the main cognitive deficits in obese pediatric patients.

## 1. Introduction

Obesity is defined as an abnormal accumulation of fat, with a 20% increase in excess weight relative to height and age in the case of pediatric obesity [1]. In Mexico, according to the results of the National Health and Nutrition Survey 2018, a prevalence of 20% pediatric obesity was reported in children aged 5–11 years [2], making it a country with one of the highest rates of pediatric obesity in the world [3]. Several parameters are used to classify obesity in children and adolescents; according to the World Health Organization, a weight for height ≥3 standard deviations above the median of child growth standards is considered, whereas the Centers for Disease Control and Prevention and the Endocrine Society define obesity as a body mass index (BMI) ≥ 95th [4,5]. By obtaining the percentile and making a diagnosis, pediatric obesity can be classified as either endogenous organic, exogenous, or nutritional [6,7]. The International Classification of Diseases (ICD)-10 classifies obesity as obesity due to excess calories (overnutrition), whereas the ICD-11 classifies it as obesity, overweight, adiposity, or excess of certain nutrients [8,9].

Increased BMI has been associated with decreased gray matter volume in the left orbitofrontal cortex, right inferior frontal gyrus, and right prefrontal cortex [10,11,12,13,14]. In addition, volumetric differences have been reported in the parahippocampal gyrus, fusiform gyrus, and lingual gyrus, as well as increases in white matter volume in the frontal, temporal, and parietal lobes [15,16]. With respect to the functional areas affected by obesity, cognitive deficits can be associated with obesity in processes such as attention, executive functions, visuomotor processing, and motor skills [17,18,19,20]. These functions are extremely important for cognitive neurodevelopment, as one function precedes the other during this dynamic growth process; therefore, low processing in attention and memory affects the development of more complex functions, such as executive functions, which may lead to future learning disorders [21,22,23].

As an alternative to neurodevelopmental deficits, cognitive habilitation or neurohabilitation has been proposed [24]. Its aim is to reactivate neurocognitive functions through frequent external exercises, which can be carried out via different tools, such as computerized, noncomputerized, or mixed methods, which have shown significant results [25,26,27]. Within the group of digital alternatives, there are mobile neurocognitive training applications (games that progressively increase the difficulty of the exercises and are easily accessible to subjects), such as Peak, which is a brain training app that helps improve cognitive skills, such as attention and memory, focus, and problem solving [28,29]. Peak (Synaptic Labs) is a new app of brain games developed by neuroscience, cognitive science, and education specialists from the University of Cambridge, King’s College London, UCL, and Yale School of Medicine. Peak is essentially a free app with more than 40 downloadable games (including the memory game Wizard, the attention game Decoder, the decision-making game Connect Em Up, and the game Handling Emotion) designed to train and develop attention, memory, language, reasoning, executive function, emotion processing, and perception [30,31]. In addition, this application has a personal coach to customize the appropriate training for the individual to maximize skill improvement and monitor progress [30]. Peak started with the development of “Decoder” a novel game to develop visual attention being tested on healthy young adults and with memory game “Wizard” used in patients with schizophrenia [32,33]. To date, there are no studies that have demonstrated the effectiveness of the application in patients with obesity, and since it is a worrying health problem in Mexico and the world that causes neurophysiological changes in the individual, we suggest that the application would help learning and improve the executive functions of children with this condition in addition to initiating research on this new application in pediatric patients.

Few studies have been conducted in this area, but those that have been reported have shown improvements, particularly in attention and memory [34,35,36]. In school-aged pediatric patients, significant results have been obtained with a system for habilitating attention and memory through a computerized system, revealing an increase in these cognitive processes [37,38]. However, the Peak app is a more complete digital technology that addresses a greater number of cognitive skills; it is translated into Spanish to be more accessible in Spanish-speaking countries; the games range from simple to complex; the interface is easy to use, with instructions accessible to all types of users; and its development is based on neuroscience [39]. Accordingly, the aim of this work was to measure, for the first time, cognitive deficits and the effects of Peak digital cognitive neurohabilitation therapy in patients diagnosed with exogenous or nutritional obesity.

## 2. Materials and Methods

### 2.1. Study Design and Ethical Considerations

A case series was evaluated before and after an intervention using the Peak app in pediatric patients with obesity at the Neurohabilitation and Behavior Unit of the National Institute of Pediatrics (INP). Patients were recruited from October 2022 to October 2023. Participation was strictly voluntary. Parents or guardians and patients were informed of this study’s objectives, methods, benefits, risks, and drawbacks. Written informed consent was subsequently obtained from each patient. The INP’s Research and Ethics Committees approved the protocol (registration number 2022/058). Throughout this study, the guidelines of the institutional research committee and the code of ethics of the Declaration of Helsinki were followed.

### 2.2. Participants

Patients who were diagnosed with pediatric obesity and who had complete electronic medical records were included. The diagnosis was made by the Obesity Clinic Service of the INP. Patients were between 6 and 18 years of age. Once the parents and patients agreed to participate, they were explained how to use the Peak app, and the children then underwent a pretest neuropsychological assessment. Six months later, after completing the series of interventions, a posttest was conducted. Of the 40 patients who agreed to participate, only 7 completed this study.

Notably, patients with cognitive deficits associated with another disease or with any relevant psychiatric or psychological disorder were not included. Parents and patients were free to withdraw from this study at any time. Patients who missed more than four follow-up appointments and those who did not complete rehabilitation treatment with the app were excluded. The evaluations and supervision of the interventions were conducted by neuropsychologists trained in the use and management of the Peak attention and memory app.

### 2.3. Instruments

#### 2.3.1. Socioeconomic Identification Card

The Socioeconomic Identification Card of the National Institute of Pediatrics includes the general data of the patient and parents, including income and expenses, housing, place of residence, state of health of the family, and diagnosis. Five indicators are considered with percentage values, and together, they constitute 100% of the classification (family income, nutrition, type of housing, place of origin, and health status). The patients were categorized into Level 1x (extremely low), Level 1 (very low), Level 2 (low), Level 3 (medium low), Level 4 (medium high), Level 5 (high), Level 6 (very high), and Level K (extremely high) [40].

#### 2.3.2. Neuropsychological Attention and Memory Battery (NEUROPSI)

NEUROPSI is a neuropsychological test that examines cognitive processes related to attention and memory. NEUROPSI was developed and evaluated in Mexico. It includes subtests for orientation, attention, memory, language, visuospatiality, visual perception, and executive functions, which together assess overall cerebral cognitive functioning. The test consists of simple and short items administered according to the total number of subscales, taking approximately 45–50 min to complete. Data from the subtests are converted into standardized scores with a mean of 10 and a standard deviation of 3, providing high reliability and a Cronbach’s alpha greater than 0.80 in the Mexican population aged 6 years and older. Interpretation of the standardized scores for each area allows for the classification of an individual’s performance as follows: high normal, 116 and above; normal, 85–115; mild to moderate impairments, 70–84; and severe impairments, less than 69 [41].

#### 2.3.3. Digital Peak App

The intervention was carried out through the Peak mobile application (Brain Games; Synaptic Labs) [29], which was developed by experts in neuroscience, cognitive science, and education as a means of brain training. It features over 45 brain games that focus on memory, attention, problem solving, and mental agility, among other cognitive processes. Eight games were selected, four of which focused on attention and four on memory. When a game was selected, a tutorial with instructions was displayed; at the end, the performance results were presented. The selected games were divided into weeks, as shown in Table 1.

### 2.4. General Procedure

First, the relevant clinical and demographic data were recorded, and then an initial assessment was conducted to determine the degree of cognitive impairment via the NEUROPSI test. This assessment was conducted in two sessions (one per week), each lasting 60 min. The patient was subsequently monitored once a week (60 min), with the aim of evaluating weekly use and complications during the use of the application by the patient and the caregiver or family member, who supervised the use of digital games using the Peak app, lasting 20 min, five times a day. Finally, a postintervention evaluation was performed.

### 2.5. Statistical Analysis

Statistics were reported for the following variables: sex, age, scholar grade, socioeconomic level, and BMI. The means ± standard deviations (SD) of the pretest and posttest values of the different cognitive processes and subprocesses were calculated. For the comparison of means, the *t* test for paired samples was used, which is based on the comparative proposal for this type of study for psychological interventions [42].

## 3. Results

### 3.1. Descriptive Analysis of the Population

The descriptive characteristics of the pediatric obese patients are shown in Table 2. A total of seven patients who used the Peak app were analyzed. The ages in the group ranged from 6 to 13 years, with a median age of 10 years (interquartile range: 9–12). In addition, patients were classified as nutritionally obese on the basis of their diagnosis and clinical history, such as their eating habits. Furthermore, the treatment reported by the obesity clinic in all cases was a nutritional diet and exercise on the basis of the type of obesity. Academic level was also identified: five of the patients were in primary school, and two were in high school.

### 3.2. Cognitive Evaluations of Pediatric Patients with Obesity

To compare the clinical results before and after the treatment, the natural results were obtained via subscores of the graded exercises corresponding to orientation and attention (Table 3). A significant change can be observed with the increase in the processing of the attentional in time orientation, cube proregression, and digital retention and detection.

In terms of the subscores in memory codification, there was an increase in processing for all threads, but a greater change was observed in the regressive cubes, memory curves, and associated pairs (Table 4).

In relation to Table 5, a change was obtained in the processing of most of the subprocesses, but above all, the greatest change was observed in spontaneous verbal memory, verbal memory by signals, verbal memory total recognition, evocation memory of associated pairs, and evocation of semi-complete figures.

No statistically significant changes were observed in the attentional functions related to executive processing and memory on the NEUROPSI, but clinical changes were observed in category formation, nonverbal phonological fluency, motor functions, Stroop interference, and Stroop hit (Table 6).

Finally, in relation to the total scores, statistical significance was obtained in three areas, attention and executive functions, memory, and attention with memory, and the diagnostic ranges changed from severe changes to mild changes in attention–executive functions and memory (Table 7).

## 4. Discussion

As previously described, childhood obesity is a health problem that is increasing, and Mexico is one of the countries with the highest rate of this condition [2,3]. This multifactorial disease involves a series of problems at the social and medical levels and has been shown to impact the neurodevelopment of the central nervous system and neurocognitive level [43,44]. In this series of cases where obesity was classified as exogenous or nutritional [6,7] (caused by excessive consumption of frequent calories), severe deficits in neurocognitive development were observed.

A constant negative feedback system of the hunger–satiety axis and an evaluation of the deterioration of anatomical structures such as the hypothalamus and the arcuate nucleus, which are associated with memory processing, attention, and executive functions, have been reported [45,46,47]. This agrees with the pretest results of this study, where low initial ranges of cognitive performance were obtained. Other work has shown that obesity may be associated with cognitive impairment, leading to an increase in the prevalence of dementia [48]. Metabolic syndrome and the dysregulation of insulin signaling play critical roles in glucolipotoxicity and the proinflammatory response, which causes oxidative stress [49]. Neuroinflammation induces significant changes in brain metabolism and white matter architecture, with cerebrovascular reactivity leading to cognitive impairment [50,51]. The activation of nuclear factor kappa light chain enhancer of activated B cells (NF-kB) and AMP-activated protein kinase (AMPK) has also been linked to depression and cognitive impairment [44]. Since these processes are important in the developmental sequence, if a process is not optimally consolidated in its developmental stage, it will affect the development of the most complex cognitive processes. A recent study revealed that subjects with a high BMI, corresponding to obese individuals, performed poorly on tasks related to emotional function, memory, attention, and executive function [52]. In Mexican subjects, the risk of impaired executive function was shown to increase significantly with the presence of overweight/obesity [17]. Other authors have also associated high BMI with poor cognitive performance [20,21]. Previously, we showed that, in pediatric patients with congenital heart disease, impaired executive functions can affect academic performance by reducing the ability to attend, register, encode, evoke, and abstract information, complicating the increase in cognitive capacity for each age range and school stage, as was observed in patients with a diagnosis of obesity [24,52,53].

The therapeutic alternatives proposed for cognitive interventions are very practical resources for cognitive development specialists since they can promote adherence to treatment and are perceived by patients as games that constantly and frequently activate cognitive processes, increasing the level of complexity of each level or task [30,38,54]. As in previous studies, positive changes with digital alternatives for neurohabilitation in attention and memory [24,25] were observed in posttreatment measures in our patients with obesity, generating clinically significant changes in total scores, as subprocesses of these information processing networks, after Peak application. In one study, a serious game was shown to increase cognitive restraint and reduce unhealthy food consumption [38]. Another study using a home-based iPad executive function training program in obese children reported significant improvements in attention and speed, working memory, cognitive flexibility, inhibitory control, planning and modification of food choices, BMI, and waist circumference [55]. Accordingly, the use of digital neurohabilitation training improved cognitive tasks but could also modify eating behavior [56]. Regarding the Peak app, there are no studies that have tested this application, mainly because it is a new app; however, this application was developed initially for patients with neurological or psychiatric conditions such as attention deficit hyperactivity disorder, Alzheimer’s disease, Parkinson’s disease, schizophrenia, or traumatic brain injury. There is a history of two applications that were initially tested and are now included in the app, the Wizard and Decoder games. Decoder, a cognitive training game for sustained visual attention, was tested in healthy young adults, who showed improvement in visual attention with high levels of enjoyment and motivation throughout all hours of play [32]. Another study also mentioned cognitive improvement in patients with schizophrenia using a memory game application (now included in the Peak app under the name “Wizard”). It was observed that patients improved in their episodic memory, daily functioning, and motivation [33]. Other predecessors to the Peak application were a computational model for decision tree search (included in the Peak app as “Connect ‘Em Up”) using a variant of tic-tac-toe, where it was observed that players searched more as they improved during learning and planned better sequential decisions, and another digital tool for emotions named attentional bias modification (included in the Peak app as “Handling emotion”) used in anxiety disorders [57,58]. In addition, the Peak app has been tested as a series of seven cognitive training mobile games used with healthy elderly subjects and patients with cognitive impairment to improve memory, and with computerized cognitive training used with patients with mild depressive symptoms to analyze executive function and processing speed [59,60]. As of recently, two research projects are being developed to evaluate the Peak app in relation to brain injury and the impact of workspace quality on learning [30].

Executive functions are mediated mainly by frontal brain regions, which play important roles in self-regulation; behavioral inhibition, shifting, and goal-directed behavior are important cognitive skills for weight-related behaviors [18]. The presence of obesity causes functional and structural connectivity abnormalities in the frontostriatal system, resulting in deficits in cognitive flexibility [52]. A higher BMI is associated with hippocampal atrophy, neuroinflammation, and memory deficits [21,45,61]. The use of digital technology could induce changes in frontal and hippocampal volume/connectivity and probably also lead to a decrease in neuroinflammation through the modulation of markers such as interleukin-6 and tumor necrosis factor-alpha or adipokines [19,45]. These strategies may help reduce the risk of dementia as well as neurodegenerative pathologies in the future.

Cognitive mobile games are a viable alternative for neurohabilitation, as they have demonstrated significant changes in attention networks and memory, which increases the processing of the cortices related to executive functions so that this favors the processing of complex tasks such as arithmetic calculation and reading writing, avoiding problems such as frustration and educational problems [34,54]. It has been suggested that the use of brain games is effective in training cognition before the onset of dementia or effectively improves or maintains cognitive function [35,36]. A major advantage of the Peak mobile application used in this study is that it can be used at home, between 3 and 5 times a day, after school, and by having the characteristic of digital play through an app, patients perceived it as a pleasant interface that is easy to use, and parents or guardians reported improvements in performing school and homework assignments. However, the limitations of this study were the evaluation of the effects of risk factors such as lifestyle, comorbidities, or sexual activity that could influence cognitive function.

## 5. Conclusions

In conclusion, the total sample of patients with a diagnosis of exogenous obesity resulted in a general diagnosis of severe cognitive deficits, especially in the attentional networks and memory and executive functions. When the digital intervention Peak was implemented to enable cognitive processing, improvements in different subprocesses, such as total processes for improvement and use of the neurocognitive system, were observed. However, due to the limited number of patients, these observations must be treated with caution. Future studies should be conducted in habilitation patients with endogenous obesity and also in patients with other cognitive impairment conditions, such as neurological and psychiatric diseases, to evaluate the effectiveness of this digital tool in learning and improving executive functions.

## Figures and Tables

**Table 1 brainsci-15-00353-t001:** Division of sessions for week and games.

Weeks	Attention Games	Memory Games
1 to 4	Object Find	Perilous Path
5 to 10	Decoder	Spin Circle
11 to 16	Rush Back	Partial Math
17 to 24	Must Sort	Wizard

**Table 2 brainsci-15-00353-t002:** Characteristics of pediatric patients with obesity.

Variable	Characteristics	Values
Sex ^1^	Male	6 (85.7%)
	Female	1 (14.3%)
Age ^2^	Years	10.00 (9–12)
Scholar grade ^1^	1–3° primary	2 (28.6%)
	4–6° primary	3 (2.9%)
	1–2° high school	2 (28.6%)
Socioeconomic level 1 ^1^	Level 1	3 (45%)
	Level 2	4 (55%)
BMI ^2^	Obesity nutritional/ exogenous	28.79 (24.76–34)

^1^ Counts and percentages. ^2^ Median and interquartile range.

**Table 3 brainsci-15-00353-t003:** NEUROPSI pretest and posttest subscores in terms of orientation and attention.

Variable	Mean			
Subprocess	Pretest	Posttest	*t*	*p* Value
Orientation in person	0.71 ± 0.48	0.71 ± 0.48	−0.487	0.00
Orientation in time	2.57 ± 1.27	**3.28 ± 1.25**	−2.500	0.047 *
Orientation in space	1.14 ± 0.48	1.29 ± 0.48	−0.548	0.604
Digital retention	3 ± 1.82	**4 ± 2**	−2.646	0.038 *
Progression cubes	3 ± 2	**4.42 ± 2.29**	−2.335	0.031 *
Visual detection	6.71 ± 5.4	**10.28 ± 7.4**	−2.813	0.086
Digital detection	5.14 ± 3.1	**6.85 ± 3.4**	−2.048	0.040 *
Successive series	0.51 ± 1.13	**0.71 ± 1.11**	−1.000	0.356

The numbers in bold indicate when the patient’s clinical diagnosis changed; mean ± SD; *t* test for related samples; * *p* ≤ 0.05.

**Table 4 brainsci-15-00353-t004:** NEUROPSI pretest and posttest subscores in codification memory.

Variable	Mean			
Subprocess	Pretest	Posttest	*t*	*p* Value
Detection in digital progression	2.00 ± 1.0	**2.4 ± 1.27**	−1.441	0.200
Cube regression	3.00 ± 1.9	**3.7 ± 1.8**	−2.500	0.047 *
Memory curve	5.50 ± 1.5	**6.8 ± 0.98**	−4.000	0.010 *
Associated pairs	6.57 ± 2.29	**8.2 ± 1.88**	−6.000	0.001 *
Historical logical memory	6.57 ± 1.8	**8.00 ± 2.1**	−1.987	0.094
Thematic logical memory	3.8 ± 1.06	**4.00 ± 1.8**	−0.311	0.766
Semi-complete figure codification	12.5 ± 10.22	**13.5 ± 9.19**	−1.066	0.327
Face recognition	1.4 ± 1.5	**2.0 ± 1.41**	−1.922	0.103

The numbers in bold indicate when the patient’s clinical diagnosis changed; mean ± SD; *t* test for related samples; * *p* ≤ 0.05.

**Table 5 brainsci-15-00353-t005:** The NEUROPSI pretest and posttest subscores in evocation memory.

Variable	Mean			
	Pretest	Posttest	*t*	*p* Value
Spontaneous verbal memory	4.5 ± 2.3	**6.28 ± 3.3**	−3.618	0.011 *
Verbal memory by cues	4.42 ± 2.50	**6.42 ± 3.15**	−3.240	0.018 *
Verbal memory total recognition	6.5 ± 3.2	**9.0 ± 2.1**	−5.050	0.002 *
Associated pair evocation memory	7.0 ± 2.6	**9.0 ± 2.3**	−5.292	0.002 *
Logical memory story evocation	4.8 ± 2.8	**6.42 ± 2.59**	−1.213	0.271
Logical memory theme evocation	3.7 ± 1.6	**4.0 ± 1.8**	−0.795	0.457
Semi-complete figure evocation	7.4 ± 5.77	**9.57 ± 7.46**	−2.423	0.052 *
Names	1.0 ± 1.8	**1.7 ± 2.8**	−1.698	0.140
Face recognition evocation	1.0 ± 1.0	0.85 ± 0.89	1.000	0.356

The numbers in bold indicate when the patient’s clinical diagnosis changed; mean ± SD; *t* test for related samples; * *p* ≤ 0.05.

**Table 6 brainsci-15-00353-t006:** NEUROPSI scores for executive functions correlated with attention and memory networks.

Variable	Mean			
	Pretest	Posttest	*t*	*p* Value
Category formation	6.57 ± 4.8	**8.2 ± 5.5**	−1.816	0.119
Semantic verbal fluency	1.57 ± 0.78	1.57 ± 0.53	0.000	1.000
Phonological verbal fluency	1.28 ± 0.48	1.28 ± 0.95	0.000	1.000
Phonological nonverbal fluency	1.28 ± 1.25	**1.42 ± 1.27**	−1.000	0.356
Motor functions	12.00 ± 5.8	**14.00 ± 6.9**	−1.833	0.116
Stroop interference	1.42 ± 0.9	**1.8 ± 1.2**	−2.121	0.078
Stroop hit	1.28 ± 0.95	**1.4 ± 1.13**	−1.000	0.356

The numbers in bold indicate when the patient’s clinical diagnosis changed; mean ± SD; *t* test for related samples.

**Table 7 brainsci-15-00353-t007:** NEUROPSI total scores pretest and posttest for attention with executive functions, memory, and attention and memory.

Variable	Mean			
	Pretest	Posttest	*t*	*p* Value
Total attention and executive functions	51.71 ± 14.78	**72.18 ± 18.61**	−3.000	0.024 *
Total memory	56.57 ± 12.10	**74.28 ± 19.00**	−3.447	0.014 *
Total attention and memory	52.14 ± 10.77	**69.57 ± 18.07**	−3.177	0.019 *

High–normal = ≥116; normal = 85–115; mild–moderate alterations = 70–84; severe alterations = ≤69. The numbers in bold indicate when the patient’s clinical diagnosis changed; mean ± SD; *t* test for related samples; * *p* ≤ 0.05.

## Data Availability

The data presented in this study are available upon request from the corresponding author. The data are not publicly available due to privacy concerns.

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
