# Peer review of "Effects of Digital Neurohabilitation on Attention and Memory in Patients with a Diagnosis of Pediatric Obesity: Case Series"

_brainsci, 2025, doi:10.3390/brainsci15040353_

Round 1
Reviewer 1 Report
Comments and Suggestions for Authors
The authors presented a novel perspective on the impact of digital neurorehabilitation through the PEAK mobile app on cognitive function, including attention and memory, in children with pediatric obesity. The study involved a group of seven patients who utilized the app for six months. The results indicate significant improvements in cognitive function, suggesting that digital technologies may effectively support neurorehabilitation processes in children with obesity.
This work addresses the critical issue of obesity as a global health concern that affects metabolic processes and the normal functioning of the CNS, particularly the grey matter. The incorporation of technology (PEAK mobile app) in therapy offers a means of digital neurorehabilitation that is easily accessible to the population.
The findings imply that PEAK can enhance cognitive function, representing an important advancement in developing therapeutic approaches. However, due to the small sample size (seven participants out of 40 involved in the study), the absence of a control group, the relatively short duration of the study, and, most significantly, the lack of consideration for environmental factors (such as lifestyle, diet, and physical activity), the conclusions should be approached with caution.
My suggestions for the manuscript:
A specific hypothesis for the purpose of the study.
More detailed explanation, description of the PEAK app, how it works, e.g. what specific training mechanisms the app uses, how it adjusts the difficulty level.
Also missing from the text is information on whether PEAK has been previously tested in similar clinical studies
It is also not specified why PEAK was chosen over other neurorehabilitation apps
Author Response
Comment:
The authors presented a novel perspective on the impact of digital neurorehabilitation through the PEAK mobile app on cognitive function, including attention and memory, in children with pediatric obesity. The study involved a group of seven patients who utilized the app for six months. The results indicate significant improvements in cognitive function, suggesting that digital technologies may effectively support neurorehabilitation processes in children with obesity.
This work addresses the critical issue of obesity as a global health concern that affects metabolic processes and the normal functioning of the CNS, particularly the grey matter. The incorporation of technology (PEAK mobile app) in therapy offers a means of digital neurorehabilitation that is easily accessible to the population.
The findings imply that PEAK can enhance cognitive function, representing an important advancement in developing therapeutic approaches. However, due to the small sample size (seven participants out of 40 involved in the study), the absence of a control group, the relatively short duration of the study, and, most significantly, the lack of consideration for environmental factors (such as lifestyle, diet, and physical activity), the conclusions should be approached with caution.
Response:
Thank you for your kind comment. We changed the conclusions.
Comment:
My suggestions for the manuscript:
A specific hypothesis for the purpose of the study.
Response:
The hypothesis has been added to the Introduction section of the manuscript.
Comment:
More detailed explanation, description of the PEAK app, how it works, e.g. what specific training mechanisms the app uses, how it adjusts the difficulty level.
Also missing from the text is information on whether PEAK has been previously tested in similar clinical studies
Response:
The description of Peak's application has been placed in the Introduction section of the manuscript. Information about the digital predecessors of this new application has been added to the Discussion section as well as new projects. It should be mentioned that as this is a new technological application, there is not much scientific information about it yet.
Comment
It is also not specified why PEAK was chosen over other neurorehabilitation apps
Response:
The Peak application is the most complete and novel digital technology for assessing a complete set of cognitive abilities, and it is an application developed on the basis of other applications previously studied in healthy subjects and in some other pathologies that cause cognitive impairment, and it is an application that continues to be studied in other research projects, some of which have already been published in prestigious scientific journals. This information has been added to the Introduction section of the manuscript.
The main changes are highlighted in yellow
Reviewer 2 Report
Comments and Suggestions for Authors
The article is extremely relevant and innovative, as it presents a therapeutic proposal to treat cognitive deficits in obese children. However, in order for it to be in an acceptable format for publication, the authors must clarify some specific points.
1. Previous studies have already related the tool known as PEAK. In this article, the authors must describe the meaning of the acronym "PEAK" so that we can compare it with previous studies. In addition, this method has apparently been tested on individuals with other diagnoses, such as ASD (autism), for example. Would they have similar mechanisms? Is there any advantage between them, if they are different? It is worth including this subject in the discussion, if they are similar.
2. The PEAK treatment lasted 6 months, including the neurocognitive evaluation. Why 6 months? Is there any reference or previous study that guarantees this time?
3. After this 6-month period, were the patients monitored? Did they maintain a lower degree of diagnosis or did they regress again?
4. The patients used PEAK 5 times a day for 20 minutes. Why this interval? How did you ensure that the schedules were met? This information should be included in the methodology.
5. The diagnostic interval increased from severe to moderate. To clarify whether the effects were immediate or delayed, an analysis could be performed over 3 months. It is also important to know whether there was regression to the previous state after the period without treatment?
6. The authors concluded that the intervention with the PEAK application improved attention and memory. However, only 7 children were tested. A low n is understandable in studies with humans, but possible variations may make such conclusions unfeasible, due to the small number of cases studied. More obese pediatric patients need to be included in these tests.
7. Were the patients monitored from a clinical perspective during this period? For example, were laboratory tests performed? Was there any information about blood biochemistry, hepatic steatosis or any metabolic changes in these patients during the tests?
Author Response
Comment:
The article is extremely relevant and innovative, as it presents a therapeutic proposal to treat cognitive deficits in obese children. However, in order for it to be in an acceptable format for publication, the authors must clarify some specific points.
Response:
Thank you for your kind opinion
Comment:
- Previous studies have already related the tool known as PEAK. In this article, the authors must describe the meaning of the acronym "PEAK" so that we can compare it with previous studies. In addition, this method has apparently been tested on individuals with other diagnoses, such as ASD (autism), for example. Would they have similar mechanisms? Is there any advantage between them, if they are different? It is worth including this subject in the discussion, if they are similar.
Response:
Peak is the name of the app, it has no acronyms. The game was developed for the study by researchers at the Department of Psychiatry, University of Cambridge. Peak was then licensed and made into a more accessible mobile game. The description of Peak's development was included in the Introduction section of the manuscript.
These are some references to digital applications or games prior or posterior to the Peak app that have been used in healthy people or people with schizophrenia, depression, mental aging, and cognitive impairment.
Savulich G et al. (2019) Improvements in Attention Following Cognitive Training With the Novel “Decoder” Game on an iPad. Frontiers of Behavioural Neuroscience. 21;13:2.
Opheusden B, Galbiati G, Bnaya Z, Li Y, Ma WJ (2017). A computational model for decision tree search Proceedings of the Cognitive Science Society. Mind Modeling: 1254-1259.
Bonnechere, B. et al (2018) ‘The use of mobile games to assess cognitive function of elderly with and without cognitive impairment’, Journal of Alzheimers Disease, 64(4), pp1285-1293
Motter, J.N. et al (2019) ‘Computerized cognitive training in young adults with depressive symptoms: Effects on mood, cognition, and everyday functioning’, Journal of Affective Disorders, 245, pp 28-37.
We should mention that these references were added in the Discussion section of the manuscript.
As for the mechanisms, they would be different since the etiology of ASD is a deficit in the ability to recognize and internalize emotions in others, which corresponds to the prefrontal and theory of mind circuitry. On the other hand, in obesity, alterations at the CNS level correspond to the memory network, which alters the rest of the circuitry. It could be interesting to replicate this in other pathologies such as ASD, ADHD, or any neuropsychological syndrome that may affect cognitive development.
Comment:
- The PEAK treatment lasted 6 months, including the neurocognitive evaluation. Why 6 months? Is there any reference or previous study that guarantees this time?
Response:
It was programmed for 6 months for two reasons:
-The first is that, according to the principles of cortical plasticity, irradiation and cellular concentration, the stimulus should be constant over a long period of time in order to have a greater effect on the acquisition of the skills to be reinforced.
-The second is that according to the principles of neuropsychological evaluation, it is suggested to make the evaluation after 6 months, to avoid the learning bias of the neuropsychological test, this principle changes for the development due to the deficit that may exist, which incapacitates the subject to encode and evoke, however, we observed in the study that the patients found it really complicated to progress from simple to complex exercises, and with this space of time we allowed them to progress in their interventions.
This study was carried out in the same way:
Zaldumbide-Alcocer FL, Labra-Ruiz NA, Carbó-Godinez AA, Ruíz-García M, Mendoza-Torreblanca JG, Naranjo-Albarrán L, et al. Neurohabilitation of cognitive functions in paediatric epilepsy patients through LEGO®-based therapy. Brain Sciences [Internet]. 2024 Jul 1;14(7):702. Available from: https://www.mdpi.com/2076-3425/14/7/702
Comment:
- After this 6-month period, were the patients monitored? Did they maintain a lower degree of diagnosis or did they regress again?
Response:
They have been contacted for follow-up and some attend the evaluation appointments, but some have not been able to attend due to distance or economic reasons. After an intervention at the neurohabilitation unit, they are followed up monthly, bimonthly, quarterly, half-yearly and annually.
Comment:
- The patients used PEAK 5 times a day for 20 minutes. Why this interval? How did you ensure that the schedules were met? This information should be included in the methodology.
Response:
This was reported by the parents at the follow-up session. This information is added to the article.
Comment
- The diagnostic interval increased from severe to moderate. To clarify whether the effects were immediate or delayed, an analysis could be performed over 3 months. It is also important to know whether there was regression to the previous state after the period without treatment?
Response:
In this type of case, it is recommended to do an intermediate evaluation, but we decided to wait, in order to avoid any kind of bias in the learning of the neuropsychological battery, and also to do an evaluation every year to observe if the changes are maintained. This is a good observation for future studies.
Comment:
- The authors concluded that the intervention with the PEAK application improved attention and memory. However, only 7 children were tested. A low n is understandable in studies with humans, but possible variations may make such conclusions unfeasible, due to the small number of cases studied. More obese pediatric patients need to be included in these tests.
Response:
Of course, the study started with a database of more than 40 subjects, 10 were left and later they dropped out, but one disadvantage of the institute is that it is far away from the patients and what the parents told us is that they have to attend many sessions, which is sometimes difficult for them financially, another disadvantage is that they have to miss school lessons because the appointment was in the morning.
Comment:
- Were the patients monitored from a clinical perspective during this period? For example, were laboratory tests performed? Was there any information about blood biochemistry, hepatic steatosis or any metabolic changes in these patients during the tests?
Response:
It was not included because the aim was not to correlate with any other biochemical process, but only with clinical scores because the aim was to evaluate the presence of dysmnesia (memory deficit) or cognitive impairment in obesity. In the Obesity Clinic, the children are diagnosed and classified according to their examinations and laboratories, and they are classified according to the type of obesity, which in this case is exogenous due to diet and sedentary lifestyle. In the future, however, it will be considered to analyse some markers in patients with endogenous obesity, which can give more information about the alteration of the nervous system.
The main changes are highlighted in yellow in the manuscript
Round 2
Reviewer 1 Report
Comments and Suggestions for Authors
I would like to thank the authors for their contributions to the revision of the manuscript. Most of my previous comments have been taken into account.